# A Study of Nano-Tungsten Colloid Preparing by the Electrical Spark Discharge Method

**DOI:** 10.3390/mi13112009

**Published:** 2022-11-18

**Authors:** Chaur-Yang Chang, Kuo-Hsiung Tseng, Jui-Tsun Chang, Meng-Yun Chung, Zih-Yuan Lin

**Affiliations:** Department of Electrical Engineering, National Taipei University of Technology, Taipei 10608, Taiwan

**Keywords:** electrical spark discharge method, nano-W colloid, electrical discharge machining, interelectrode gap

## Abstract

This study developed an energy-enhanced (ee)-micro-electric discharge machining (EDM) system for preparing nano-tungsten (nano-W) colloids. This system enables spark discharge using tungsten wires immersed in deionized water, to produce nano-W colloids. Compared with the chemical preparation method, the processing environment for preparing colloids in this study prevented nanoparticle escape. Among the nano-W colloids prepared using the ee-micro-EDM system and an industrial EDM system, the colloid prepared by the ee-micro-EDM system exhibited a more favorable absorbance, suspensibility, and particle size. The colloid prepared by the ee-micro-EDM system with a pulse on time and off time of 10–10 μs had an absorbance of 0.277 at a wavelength of 315 nm, ζ potential of −64.9 mV, and an average particle size of 164.9 nm. Transmission electron microscope imaging revealed a minimum particle size of approximately 11 nm, and the X-ray diffractometer spectrum verified that the colloid contained only $W_{2.00}$ and W nanoparticles. Relative to industrial EDM applications for nano-W colloid preparation, the ee-micro-EDM system boasts a lower cost and smaller size, and produces nano-W colloids with superior performance. These advantages contribute to the competitiveness of the electrical spark discharge method in the preparation of high-quality nano-W colloids.

## 1. Introduction

Tungsten (W) is a transition metal with high hardness, a high melting point, and high heat conductivity. The metal has high stability and exists in various compounds in nature, with tungsten carbide (WC) and tungsten trioxide (WO_3_) the most common forms. Research has been performed on WC applications in mechanical processing [1,2] and catalysts [3,4], as well as on its material properties [5,6]. Studies have also been conducted on the material properties of WO_3_ [7,8] and its applications in thin films [9,10] and sensors [11,12]. Research topics explored regarding the development of tungsten nanocomposite materials include the mechanical alloying of copper–tungsten (C-W) composite alloys [13], friction stir alloying of aluminum–tungsten (Al-W) composite [14], and discharge plasma sintering of copper–tungsten nanocomposite materials [15]. Given the widespread application of nano-tungsten (nano-W) materials, their production is crucial for nanomaterial research. Currently, chemical methods are generally employed for the preparation of nano-W materials. However, such methods risk contamination and are plagued with nanoparticle dissipation problems. The literature indicates that exposure to nanoparticles can have adverse health effects. For example, the inhalation of nanoparticles may result in lung tissue lesions [16]. Therefore, resolving the problem of flying dust in the preparation environment is crucial [17]. The use of electrical discharge machining (EDM) to obtain nanomaterials from melting the surface of electrode materials is called the electrical spark discharge method (ESDM) [18,19,20]. One of the characteristics of ESDM preparation of nano-colloids is that the process of colloid generation and collection is completed in the dielectric liquid, so the nanoparticles do not dissipate in the process environment. A recent study employed an ESDM for the preparation of ${\mathrm{WO}}_{3}$·$H_{2}O$ nanoparticles [21]. The study adopted pulse generators, vibrators, deionized water, and two W electrodes to prepare nanoparticles in deionized water, thereby avoiding problems encountered in the chemical method. The proposed nano-W colloid preparation method has a simple equipment framework, but it lacks the server-based interelectrode gap (IEG) control mechanism of industrial EDM. Therefore, the IEG cannot be dynamically adjusted in accordance with the IEG resistance and often exhibits poor discharge performance [22]. This reduces the nanoparticle yield per units of time and electrode material. The ESDM in industrial nano-W colloid preparation entails large, costly equipment. To avoid the aforementioned disadvantages, this study redesigned the servo circuit of an existing micro-EDM system used for nano-W colloid preparation. In this paper, the existing micro-electrical discharge machining systems are referred as the existing-micro-EDM system [23]. Compared with the traditional EDM, our system used two tungsten alloy wires with a diameter of 1 mm and a purity of 99.97% as the electrode material. These two tungsten alloy wire electrodes were used to replace the electrodes and workpieces of traditional EDM. When the distance between the two tungsten alloy wire electrodes was kept within a few microns, the pulse voltage of the system made the electrode gap show good spark discharge characteristics. This study evaluated the advantages of the energy-enhanced (ee) ee-micro-EDM system in nano-W colloid preparation, by assessing the properties of nano-W colloids prepared using the ee-micro-EDM system and an industrial EDM system.

## 2. Materials and Methods

### 2.1. ESDM Principle

The principle of ESDM is to activate periodic spark discharges in the dielectric fluid (DF) in the electrode gap using a pulse voltage. This subjects the surface material of the electrodes to high-temperature vaporization and rapid condensation in the DF, thus producing nanoparticles. The preparation efficiency is associated with the IEG discharge rate. Factors influencing the IEG discharge rate include the pulse voltage, time variables, slag residue between the IEG, and the dielectric strength of the DF. Figure 1 presents the discharging process of a single cycle, and the discharged voltage and current. The anode and cathode of the pulse voltages are separately connected to the two electrodes. $V_{\mathrm{IEG}}$ and $I_{\mathrm{IEG}}$ represent the IEG voltage and current, respectively [24]. Figure 1a displays the sparking process of the ESDM, with $T_{1}$, $T_{2}$, $T_{3}$, and $T_{4}$ sequentially representing the discharge, ionization, slag generation and expulsion, and insulation recovery of the DF stages, respectively. $T_{1}$–$T_{3}$ occur during the $T_{\mathrm{on}}$ stage of the pulse period; $T_{4}$ occurs during the $T_{\mathrm{off}}$ stage of this period. Figure 1b indicates $V_{\mathrm{IEG}}$ and $I_{\mathrm{IEG}}$ during $T_{1}$–$T_{4}$. In the electrode gaps in the T_1_, T_2_, T_3_, and T_4_ periods of Figure 1, the discharge phenomenon, voltage, and current are described as follows: During the $T_{1}$ stage, few DF molecules in the IEG are ionized in the electric field. A discharge channel is formed in the IEG, and trace amounts of electrons are emitted from the cathode surface to the anode surface [25,26]. During the subsequent $T_{2}$ stage, large numbers of DF molecules in the IEG are subjected to high-speed impact by electrons and are ionized. The aforementioned process of ionization by collision rapidly increases the number of positive ions and free electrons in the IEG, both of which rapidly shoot toward the opposite electrode, forming a narrow plasma column in the IEG. The resistance of the DF ($R_{\mathrm{df}}$) rapidly drops due to insulation breakdown, which results in a decrease in $V_{\mathrm{IEG}}$ and a rapid increase of $I_{\mathrm{IEG}}\;$to the maximum value. During the $T_{3}$ stage, the numerous free electrons and positive ions in the IEG rapidly impact the opposite electrode under the effect of the electric field. On impact, the kinetic energy is converted to heat, which melts and vaporizes the surface material of the electrodes and causes the DF in the IEG to vaporize and expand [27,28]. The impact caused by the vaporization and expansion of DF pushes the metallic vapor away from the IEG, where it makes direct contact with the surrounding low-temperature DF and rapidly condenses into nanoparticles or submicron particles. At this stage, $I_{\mathrm{IEG}}$ is maintained at the maximum value, due to the insulation breakdown of the DF. Contrarily, $V_{\mathrm{IEG}}$ is maintained at a fixed value, as sparking continues in the IEG. During the $T_{4}$ stage, the pulse source is in the off state, and the melted material residues in the IEG are expulsed from it. The $R_{\mathrm{df}}$ returns to the insulation state and $I_{\mathrm{IEG}}$ and $V_{\mathrm{IEG}}$ rapidly drop to zero. Through the aforementioned periodic discharging process, metallic particles can be suspended in the DF.

### 2.2. Existing-Micro-EDM System

The existing-micro-EDM system is comprised of a production mechanism, a servo circuit, and a computer interface control unit (CICU). In the production mechanism, the positive and negative electrodes are immersed in deionized water in a beaker, and one of the electrodes is fixed to a slider on a slide rail. The position of the slider on the rail is controlled using a direct current motor. Therefore, the IEG is modified by controlling the motor speed. The main function of the CICU, which consists of a personal computer, VisSim software, and an RT-DAC4/PCI card [29,30], is to process the input/output signals of the servo circuit. The servo circuit is composed of a discharge circuit, a motor control feedback circuit, and a discharge state identification circuit. The function of the discharge circuit is to provide the required IEG pulse voltage, by controlling the ON/OFF state of the transistor [31,32]; the motor control feedback circuit controls the IEG and measures the IEG disposition [33]; and the discharge state identification circuit determines the IEG discharge state. When the positive and negative electrodes are kept at a small distance, the IEG pulse voltage can make the gap between the electrodes immersed in the deionized water in the beaker produce a spark discharge phenomenon.

### 2.3. Design of the Energy-Enhanced Micro-EDM System

This study used the existing-micro-EDM system for nano-W colloid preparation. However, instrument analysis revealed that the produced colloid did not possess the properties of nano-W colloid. This study inferred that the reason that the system was unable to produce nano-W colloids was because of insufficient discharge energy in the IEG. Since the discharge energy of the IEG is provided by the DC voltage source of the discharge circuit, the discharge energy is proportional to the square of the voltage. Therefore, the voltage and discharge energy of the system could be increased by redesigning the discharge circuit, thereby overcoming the problem of the insufficient discharge energy of the existing-micro-EDM system in the IEG. In the design of the system, the function of the motor control feedback unit is to capture the IEG voltage signal of the discharge circuit, to control the distance of the electrode gap. When the discharge circuit was redesigned, the motor control feedback circuit also needed to be redesigned, so that the system could properly control the electrode gap. In order to make the system have enough discharge energy to complete the preparation of nano-W colloids, this study developed an ee-micro-EDM system by redesigning the discharge circuit and motor control feedback unit. The design goal of this system was to double the voltage of the system, and to quadruple the discharge energy of the original system.

#### 2.3.1. Discharge Circuit Design

The discharge circuit of the existing-micro-EDM system is depicted in Figure 2. The circuit is composed of a 96V DC power source, an IRF740 transistor, a TLP250 optical coupled isolator, and 5 cement resistors. The rated voltage ($V_{\mathrm{DS}}$) and current ($I_{D}$) of the IRF740 transistor are 400 V and 10 A, respectively. This circuit is used to provide the high-frequency IEG pulse voltage. The voltage is generated by first using the CICU to output a PWM signal to the input end of (CON1) of the TLP250 optimal coupled insulator of the discharge circuit for the set production time ($T_{\mathrm{on}}$–$T_{\mathrm{off}}$). The TLP250 insulator subsequently outputs the signal to the IRF740 transistor gate, which results in its alteration between the ON/OFF state, in accordance with the gate signal. This rapid alteration in the ON/OFF state of the IRF740 transistor gate converts the DC power source to the high-frequency pulse voltage required for IEG discharge. When the transistor is in a conducting state, the pulse source emits a discharging current through the IEG. This circuit path is composed of serially connected resistors $R_{3}$, $R_{4A}$, and $R_{4B}$, and the electrodes. $R_{4A}$ and $R_{4B}$ serve as the current-limiting resistor of the IEG. The circuit path has an open circuit voltage of 96 V and a combined series resistance between $R_{3}$, $R_{4A}$, and $R_{4B}$ of 21 Ω. Therefore, the maximum IEG discharge current is 4.6 A. When the transistor is in the cut-off state, the energy stored in the IEG can be discharged through the circuit formed by the serial connection of $R_{2A}$, $R_{2B}$, $R_{3}$, and $R_{4A}$ with $R_{4B}$. The release of this stored energy provides the IEG with a favorable insulation property before the next discharge cycle. An LED serves as a discharge indicator; if the LED is conducted, the system is in the colloid preparation process.

To provide the ee-micro-EDM system with the ability to prepare nano-W, this study increased the power source voltage of the circuit from 96 V to 192 V, to enhance the energy discharged in the IEG. Additionally, this study redesigned the scale of specific components, particularly the three sections (i.e., A, B, and C) marked in the red dotted rectangles in Figure 2, to enable the discharging process to be conducted at a suitable range. In section A, this study replaced the 96 V direct power source with a 192 V direct power source. In section B, the resistances of $R_{4A}$ and $R_{4B}$ were increased proportionally with the increase in voltage; specifically, the two resistors were strengthened to 20 Ω. The discharge circuit of the ee-micro-EDM system has a maximum discharge current of 4.68 A. This enabled the ee-micro-EDM system to achieve the same maximum discharge current as the existing-micro-EDM system and ensured that the discharge circuit was maintained within a suitable working range. When the system voltage was increased to 192 V, the ringing effect in the discharging process generated a surge voltage greater than 500 V. This voltage exceeded the tolerance of the IRF740 transistor; therefore, a more suitable transistor was required, to ensure the operation stability of the discharge circuit. In section C, this study substituted the IRF740 transistor with an IPP60R060P7 transistor, which can withstand a voltage of 600 V. The IPP60R060P7 can ensure the stability of the discharge circuit during long-term operation.

#### 2.3.2. Motor Control Feedback Circuit Design

Figure 3 presents the motor control feedback circuit of the existing-micro-EDM system. The circuit is composed of a differential amplifier, a resistor-capacitor (RC) low-pass filter, an ISO122P isolation amplifier, a motor driver, a 74LS07 buffer, and a 6N137 optocoupler. The functions of the circuit include monitoring the IEG displacement and controlling the IEG distance. IEG displacement is monitored by first using the optical encoder mounted on the motor to output pulse signals containing data on the motor rotation speed. The CICU then converts the pulse signals into the displacement variables of the IEG. The IEG distance control function of the circuit is used to adjust the IEG distance, to ensure a suitable sparking discharge. Additionally, the circuit controls the IEG distance through the CICU, which ensures a closed-circuit IEG distance based on the feedback signals on IEG voltage. As $V_{\mathrm{IEG}}$ is a high-voltage pulse signal, the signal voltage must first be lowered and the signal converted into analog before input into the CICU. In the motor control feedback circuit, the $V_{\mathrm{IEG}}$ feedback signals must first be processed using $R_{f1}$ and $R_{f2}$, to divide the voltage. As the resistance of $R_{f2}$ is 18 times that of $R_{f1}$, $V_{\mathrm{Rf}1}$ was subjected to $\frac{1}{19^{th}}$ that of $V_{\mathrm{IEG}}$. The differential amplifier is subsequently employed to extract the voltage signal ($V_{\mathrm{Rf}1}$) in $R_{f1}$. The signal output from the differential amplifier first passes the RC low-pass filter for conversion into analog and then is transmitted into the insolation amplifier for input into the CICU. In accordance with the analog signal on IEG voltage, the CICU activates its PID controller to output the motor rotation speed control signal ($V_{\mathrm{speed}\;\mathrm{cpmmand}}$) to the motor control feedback circuit [34,35]. In this circuit, $V_{\mathrm{speed}\;\mathrm{cpmmand}}$ is first processed using the insolation amplifier for input into the motor driver. Next, the motor driver controls the rotation speed of the DC servo motor, based on the signal input, thereby controlling the motor rotation speed to ensure a favorable IEG distance for sparking discharge.

The motor control feedback circuit of the ee-micro-EDM system was designed to readjust the resistance $R_{f2}$ marked by the red dotted rectangle in Figure 3. This design was intended to prevent the input voltage for the differential amplifier ($V_{\mathrm{Rf}1}$) from increasing due to system voltage changes. This design ensures that the output signals of the motor control feedback circuit of the ee-micro-EDM system are similar to those of the existing system. In Figure 3, the resistance of $V_{\mathrm{Rf}1}$ is $\frac{1}{19^{th}}$ that of $V_{\mathrm{IEG}}$. The system voltage of the ee-micro-EDM system is 192 V, which is twice that of the existing-micro-EDM. If $V_{\mathrm{Rf}1}$ in the motor control feedback circuit of the ee-micro-EDM system is $\frac{1}{38}V_{\mathrm{IEG}}$, the $V_{\mathrm{Rf}1}$ in the motor control feedback circuit of both the ee-micro-EDM system and the existing-micro-EDM system will be identical. This study increased the resistance of $R_{f2}$ (red dotted section in Figure 3) to 37 kΩ, thus enabling the resistance of $R_{f2}$ to be 37 times that of $R_{f1}$ and ensuring that $V_{\mathrm{Rf}1}$ is $\frac{1}{38}$$V_{\mathrm{IEG}}$.

### 2.4. Nanoparticle-Size Analysis

This study analyzed the particle size of colloids through transmission electron microscopy (TEM, JEM-2100F, JEOL Ltd., Tokyo, Japan) and using a Zetasizer nanosystem (Zetasizer, Nano-ZS90, Malvern Zetasizer, Worcestershire, UK). The principle of TEM is to project an accelerated and concentrated electron beam onto the sample, and then enlarge it for imaging. The image provided by this instrument is a two-dimensional image that can simultaneously observe the size and shape of the colloidal particles. The Zetasizer measures the particle size and particle size distribution of colloids using dynamic light scattering (DLS) technology, which is based on Brownian motion. This technique uses a laser source to illuminate particles. The scattered light of particles fluctuates due to Brownian motion and time changes. Since particle size affects the Brownian motion of particles, the diffusion coefficient of particles can be obtained by measuring the intensity of the dispersed light of particles undergoing Brownian motion, and the radius of particles can be calculated using the Stokes–Einstein equation. In dynamic light scattering (DLS) technology, the distribution coefficient (Particle dispersion index, PDI) indicates the degree of particle size distribution. The smaller the PDI value, the more concentrated the colloidal particle size distribution. The dynamic light scattering (DLS) technique assumes that the particles are circular and that the particle size is a single distribution state, to describe the average particle size. Therefore, the particle size distribution is concentrated very close to the ideal normal distribution, and the measurement of the average particle size will be more accurate. In dynamic light scattering technology, the intensity of scattered light is proportional to the sixth power of particle size, so particles with larger particle sizes will be given greater weight when calculating the average particle size. If the particle size distribution is very dispersed, large particles can cause a significant increase in the calculated average particle size. DLS and TEM use different principles to measure particle size, in that the DLS analyzes the particle size measured by dispersing particles in a liquid, which is the hydrodynamic size. The objects of TEM analysis are dried particles. Therefore, the particle sizes measured by these two methods will be inconsistent. In addition, particle size analysis by DLS is affected by the particle size distribution. When the colloidal particle size exhibits a monodisperse particle size distribution and all particles have the same particle size, the particle sizes measured by DLS and TEM will be similar. If the particle size distribution of the colloid is very dispersed, i.e., the PDI of the colloid is large, then the particle size analysis by DLS and TEM will be very different. In this case, the particle size measured by DLS is usually larger than that measured by TEM.

## 3. Results

The industrial EDM and ee-micro-EDM systems were used to prepare nano-W colloids under 25 °C, 1 atm. The electrode materials used were tungsten alloy wires 1 mm in diameter and with 99.97% purity. The DF used was 200 mL of deionized water. To evaluate the properties of the prepared colloids, this study adopted ultraviolet-visible spectrometry (UV-Vis, Thermo-Helios Omega, Thermo Fisher Scientific Inc., Waltham, MA, USA) and a Zetasizer, to analyze the absorbance spectrum intensity, suspension stability, and particle distribution of the colloids. Additionally, TEM was employed to observe the particle size, shape, and components.

### 3.1. Property Analysis of the Nano-W Colloids Prepared Using the Industrial EDM

Figure 4 presents the industrial EDM system used for nano-W colloid preparation. During preparation, the open circuit voltage and the peak current were 240 V and 6 A, respectively. The preparation time variables ($T_{\mathrm{on}}$–$T_{\mathrm{off}}$) were set as 10–10, 30–30, 50–50, 70–70, and 90–90 μs; the nano-W colloids prepared using these variables were labelled C-I-EDM 10-10-30 min colloid, C-I-EDM 30-30-30 min colloid, C-I-EDM 50-50-30 min colloid, C-I-EDM 70-70-30 min colloid, and C-I-EDM 90-90-30 min colloid, respectively. The preparation duration of each colloid was 30 min.

UV–Vis spectrometers are widely used in qualitative and quantitative analysis of substances. In this study, the absorption spectrum curve of the sample solution was obtained using this instrument, and the position of the maximum absorption peak (characteristic peak) of the spectrum could be used to identify the composition of the substance. Absorbance can be used as a relative indicator of colloid-containing nanoparticle concentration. Absorbance refers to the logarithm of the ratio of the incident light intensity before the light passes through the solution to the transmitted light intensity through the solution, which is a physical quantity that measures the degree of light absorption. According to the Beer–Lambert law, the absorbance of light is proportional to the concentration of the solution, so the absorbance of the colloidal UV-Vis spectrum can be used as an indicator of the concentration of nanoparticles. The higher the absorbance of the colloid, the greater the number of suspended particles contained in the colloid. In ESDM, the energy inside the electrode gap increases with the increase of the $T_{\mathrm{on}}$ period and decreases with the increase of the $T_{\mathrm{off}}$ period. If the energy remaining in the electrode gap before the end of the $T_{\mathrm{off}}$ period is too large, the electrode gap cannot be restored to the insulating state before the next cycle of discharge. The success rate is also reduced. Figure 5 presents the UV–Vis analysis results for the colloids. The C-I-EDM 50-50-30 min colloid, C-I-EDM 70-70-30 min colloid, and C-I-EDM 90-90-30 min colloid in the figure do not show characteristic peak wavelengths. The reason for this may be that the T_on_ period was longer than or equal to 50 μs, and the electrode gap could not be restored to the insulating state before the next cycle of discharge, so good spark discharge characteristics could not be obtained [36]. The C-I-EDM 10-10-30 min colloid demonstrated an absorbance peak of 0.242 at a wavelength of 316 nm, and the C-I-EDM 30-30-30 min colloid had an absorbance peak of 0.158 at a wavelength of 315 nm. In the preparation of C-I-EDM 10-10-30 min colloid and C-I-EDM 30-30-30 min colloid, the energy input to the electrode gap in the former $T_{\mathrm{on}}$ time was smaller. Therefore, this rapidly decreased the residual energy of the electrode gap and increased the insulation state of the electrode gap during T_off_ period. This result allowed the system to obtain good spark characteristics. Since the C-I-EDM 10-10-30 min colloid had better discharge characteristics, the C-I-EDM 10-10-30 min colloid had higher absorption than the C-I-EDM 30-30-30 min colloid. As Figure 5 indicates, both the C-I-EDM 10-10-30 min colloid and the C-I-EDM 30-30-30 min colloid contained nano-W particles. The ζ potential and particle size distribution, by the number of the two colloids, are presented in Figure 6 and Figure 7. The C-I-EDM 10-10-30 min colloid had a ζ potential of −61.3 mV, an average particle size of 216.3 nm, PDI of 0.842, and 100% particle size distribution of 80.51 nm at peak 1. The C-I-EDM 30-30-30 min colloid had a ζ potential of −32.6 mV, an average particle size of 252.2 nm, PDI of 0.983, and a 99.3% particle size distribution of 68.27 nm at peak 1.

Table 1 compares the properties of the C-I-EDM 10-10-30 min and C-I-EDM 30-30-30 min colloids. Analysis of the particle size distribution by number revealed that the particle sizes of both colloids met the requirement of nanometer scale. Additionally, the absolute ζ potential of both colloids exceeded 30 mV, indicating that both exhibited a favorable suspension stability. The absorbance peak and absolute ζ potential of the C-I-EDM 10-10-30 min colloid was much greater than that of the C-I-EDM 30-30-30 min colloid, thereby indicating that the C-I-EDM 10-10-30 min colloid possessed a greater nanoparticle concentration and more favorable suspension stability. Further analysis revealed that the C-I-EDM 10-10-30 min colloid exhibited a smaller average particle size than the C-I-EDM 30-30-30 min colloid. In summary, the analysis indicated that the properties of the C-I-EDM 10-10-30 min colloid were more favorable than those of the C-I-EDM 30-30-30 min colloid.

Figure 8 displays TEM images of the C-I-EDM 10-10-30 min colloid. Figure 8a presents an image recorded under ×20,000 magnification. In the magnified image, numerous black and grey spots of nano-W particles can be observed. Figure 8b depicts part of Figure 8a (indicated by the square) at ×400,000 magnification. The nano-W particles are observed as quadrangles and have a particle size of 80 nm. This finding is consistent with the measured average particle size of 80.51 at peak 1 (Figure 6b). Figure 8c presents a magnification of the image in Figure 7b (indicated by the square) at ×800,000 at a scale of 5 nm. The width of the lattice line of the nano-W particles was revealed to be 0.223 nm.

### 3.2. Property Analysis of Nano-W Colloids Prepared Using the ee-Micro-EDM System

In the preparation of nano-W colloids using the ee-micro-EDM system, the open circuit voltage and the peak current were set as 192 V and 4.7 A, respectively. This study labelled the colloid prepared using a preparation time variable ($T_{\mathrm{on}}$–$T_{\mathrm{off}}$) of 10–10 μs and preparation duration of 16 min as the C-M-EDM 10-10-16 min colloid. According to the Tyndall effect, when light irradiates the particles with a volume smaller than the wavelength of visible light, this will produce a scattering phenomenon. Figure 9 shows the colloidal solution of C-M-EDM 10-10-16 and deionized water. The figure shows that only the colloidal solution of C-M-EDM 10-10-16 has an apparent scattering trace of laser light, which confirms that the solution contains tiny particles. The UV-Vis analysis of the colloid is presented in Figure 10. The colloid exhibited an absorbance peak of 0.277 at a wavelength of 315 nm, thereby suggesting that the prepared colloid contained nano-W particles. The ζ potential and particle size distribution by number analysis of the colloid is displayed in Figure 11; the colloid had a ζ potential of −64.9 mV, a PDI of 0.589, and an average particle size of 164.9 nm. The colloid had a 100% particle size distribution of 83.41 nm at peak 1. The absolute ζ potential of the colloid was greater than 30 mV, indicating that the colloid exhibited a favorable suspension stability.

Figure 12 presents TEM images of the C-M-EDM 10-10-16 min colloid. Figure 12a depicts Site A of the colloid under ×40,000 magnification and indicates that the size of most of the nano-W particles was 11 nm. As this study did not observe an apparent lattice line after magnifying Site A, TEM images of other sites of the sample were selected. Figure 12b depicts Site B of the sample under ×10,000 magnification. In the image, a black nano-W particle is observed. Figure 12c presents a magnification of the site indicated by the rectangle in Figure 12b at ×40,000, and Figure 13 depicts a magnification of the site in Figure 12c indicated by a rectangle at ×1,000,000. The lattice line width of the nano-W particle was observed to be 0.223 nm, a finding consistent with that for the C-I-EDM 10-10-30 min colloid.

### 3.3. Component Analysis of Nano-W Colloids

This study employed an X-ray diffractometer (XRD; Panalytical, Empyrean model, Netherlands, CuKα radiation at 45 kV) to analyze the crystal structures of the nano-W particles in the colloids prepared using the industrial EDM and the ee-micro-EDM system, namely the C-I-EDM 10-10-30 min colloid and the C-M-EDM 10-10-16 min colloid. Figure 14 depicts the XRD spectrums of the colloids. Both spectrums contained $W_{2.00}$ and W diffraction peaks. The International Centre for Diffraction Data labelled $W_{2.00}$ as 96-900-8559 and W as 00-047-1319. This result indicates that both colloids contained two tungsten products with different crystal structures, namely $W_{2.00}$ and W. Figure 14 indicates that the position of the diffraction peak at the 2θ degree angle of both colloids corresponded to the same crystal orientation. Therefore, this study inferred that the colloids contained the same crystallized materials. Table 2 lists the position of the diffraction peak at the 2θ degree angle and the Miller indices of the $W_{2.00}$ and W components in the two colloids.

### 3.4. Property Comparison of Nano-W Colloids Prepared Using Industrial EDM and the ee-Micro-EDM System

This study compared the properties of the C-I-EDM 10-10-30 min colloid and the C-M-EDM 10-10-16 min colloid, to evaluate the advantages of using the micro-EDM system to prepare nano-W colloids. The items compared were the absorbance spectrum intensity, suspension stability, average size, particle size distribution by number, and surface properties (Table 3). First, regarding the particle size distribution by number, the sizes of the particles in both samples indicated they were nanoparticles. The absolute ζ potential of both colloids exceeded 30 mV, indicating a favorable suspension stability. However, the absorbance peak and absolute ζ potential of the C-M-EDM 10-10-16 min colloid were greater than those of the C-I-EDM 10-10-30 min colloid, thereby suggesting that the C-M-EDM 10-10-16 min colloid had a higher nanoparticle concentration and more favorable suspension stability. The comparison also revealed that the C-M-EDM 10-10-16 min colloid had a smaller average particle size. According to the analysis of nanoparticle concentration, suspension stability, and average particle size, the properties of the C-M-EDM 10-10-16 min colloid surpassed those of the C-I-EDM 10-10-30 min colloid. The preparation time of the C-M-EDM 10-10-16 min colloid was 16 min, and the average absorbance per minute prepared was about 0.0173 (0.277/16). The preparation time of the C-I-EDM 10-10-30 min colloid was 30 min, and the average absorbance per minute prepared was about 0.0081 (0.242/30). Due to the higher average absorbance per minute of the C-M-EDM 10-10-16 min colloid, the productivity of the number of suspended particles was higher than that of the C-I-EDM 10-10-30 min colloid. TEM analysis revealed the particle size of the C-M-EDM 10-10-16 min colloid was 11 nm; much smaller than that of the C-I-EDM 10-10-30 min colloid. Additionally, both colloids had the same lattice line width. This indicates that the nanoparticles observed in the TEM image were of the same crystal structure material. In summary, the micro-EDM system produced colloids with a higher suspension stability, nanoparticle concentration, and smaller particle size, in less time. The experimental results showed that the micro-EDM system had excellent performance in preparing nano-tungsten colloids. The reason for this may have been that the open circuit voltage and peak current of the colloids prepared by the micro-EDM system were smaller than those of the industrial EDM. The aforementioned reasons enabled the electrode gap to quickly return to a good insulating state during the $T_{\mathrm{off}}$ period of the colloid prepared by the micro-EDM system. Therefore, its $T_{\mathrm{on}}$ period could obtain good spark discharge characteristics [37,38].

## 4. Conclusions

The ee-micro-EDM system proposed in this study increased the voltage of the existing-micro-EDM system from 96 V to 192 V. This increase of the voltage can greatly increase the discharge energy, and the electrode gap can obtain a sufficiently high temperature and high pressure for the preparation of nano W colloids due to this energy. When the preparation time $T_{\mathrm{on}}$–$T_{\mathrm{off}}$ was 10-10 μs, the ee-micro-EDM system and the industrial EDM system prepared nano-W colloids with preparation durations of 16 min and 30 min, respectively. Analysis of the instruments revealed that both colloids had Nano-scale tungsten particles with good suspension. Analysis of the colloidal performance revealed that the colloid prepared by the ee-micro-EDM system exhibited a better performance in terms of absorbance, suspensibility, and particle size. These results indicate that the ee-micro-EDM system prepared nano-W colloids with a better performance, in a shorter time, than the industrial EDM system. The conclusions are as follows:Analysis using UV–Vis and a Zetasizer showed that the colloid prepared by the ee-micro-EDM system with a pulse on time and off time of 10–10 μs and a preparation time of 16 min had an absorbance of 0.277 at the wavelength of 315 nm, ζ potential of −64.9 mV, and an average particle size of 164.9 nm. TEM analysis showed that the particle size of this colloid was about 11–12 nm.Compared with the application of the industrial EDM system for nano-W colloid preparation, the ee-micro-EDM system has the advantages of a low cost, smaller size, and production of nano-W colloids with superior performance. Compared with the chemical preparation method for nano-W colloids, the ee-Micro-EDM system for preparing colloids does not permit nanoparticles to dissipate in the process environment. In summary, the ee-micro-EDM system represents an advanced technology for preparing high-quality nano-W colloids.

## Figures and Tables

**Figure 1 micromachines-13-02009-f001:**
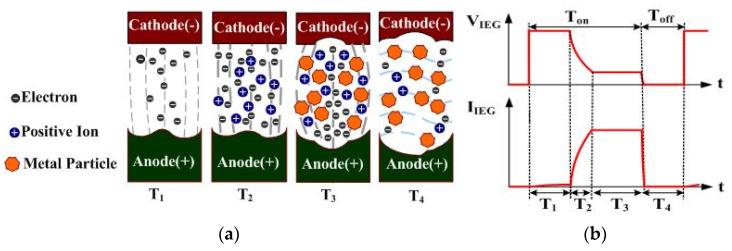
Discharge cycle of the ESDM: (**a**) discharge process; (**b**) $V_{\mathrm{IEG}}$ and $I_{\mathrm{IEG}}$.

**Figure 2 micromachines-13-02009-f002:**
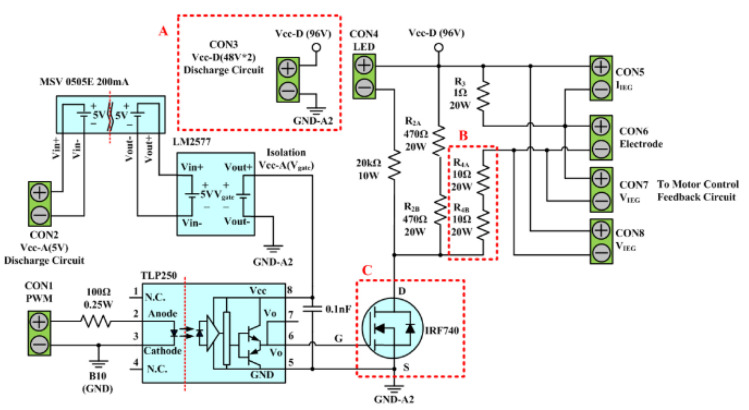
Discharge circuit of the existing-micro-EDM system.

**Figure 3 micromachines-13-02009-f003:**
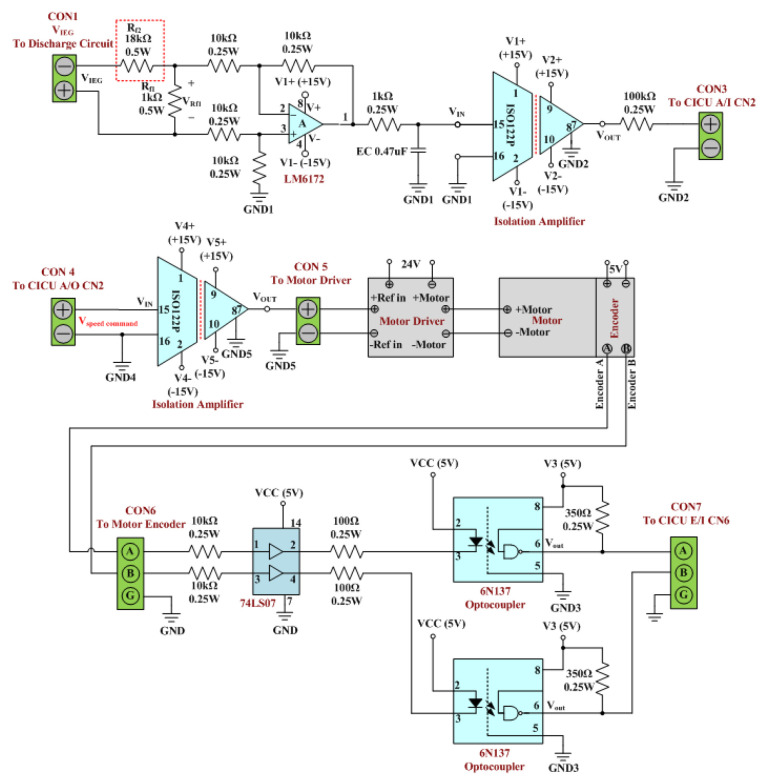
Motor control feedback circuit of the existing-micro-EDM system.

**Figure 4 micromachines-13-02009-f004:**
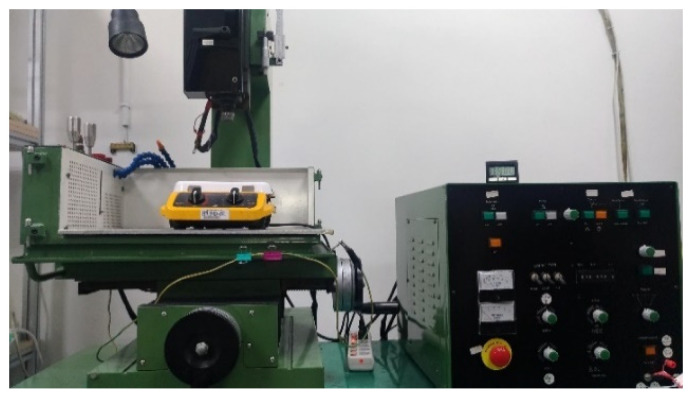
Photograph of the industrial EDM system.

**Figure 5 micromachines-13-02009-f005:**
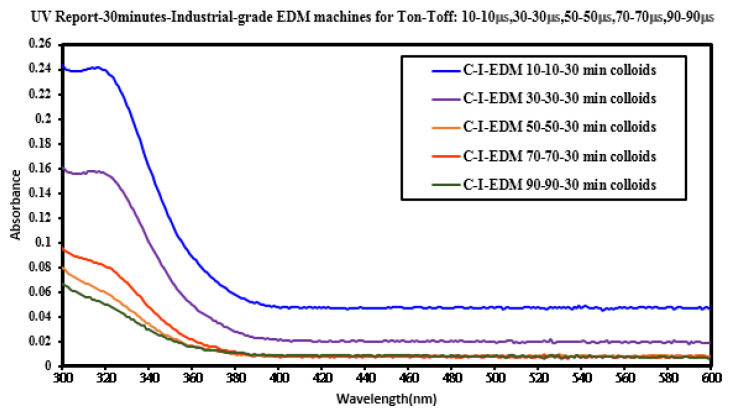
Absorbance spectrum of the five nano-W colloids prepared using industrial EDM at the five preparation times.

**Figure 6 micromachines-13-02009-f006:**
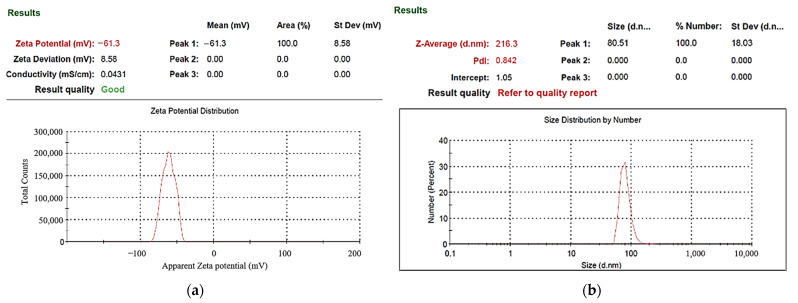
Analysis of C-I-EDM 10-10-30 min colloid: (**a**) ζ potential; (**b**) particle size distribution report by number.

**Figure 7 micromachines-13-02009-f007:**
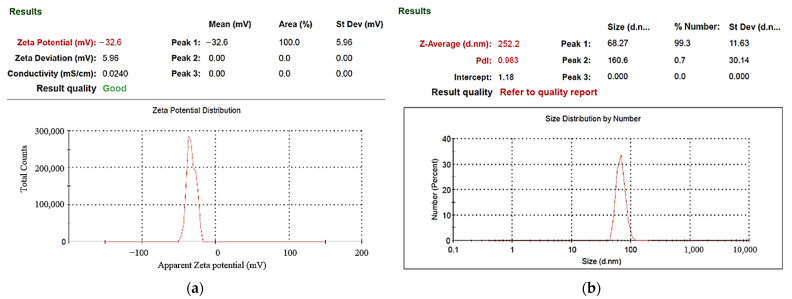
Analysis of C-I-EDM 30-30-30 min colloid: (**a**) ζ potential; (**b**) particle size distribution report by number.

**Figure 8 micromachines-13-02009-f008:**
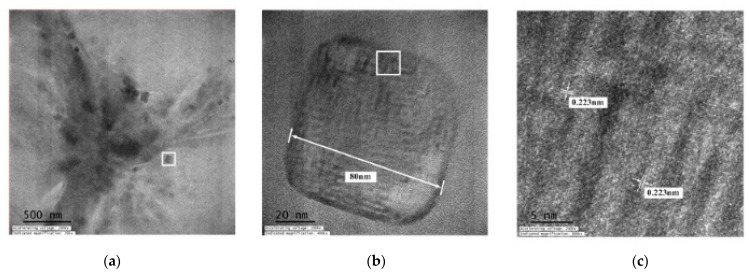
TEM images of the C-I-EDM 10-10-30 min colloid: (**a**) ×20,000; (**b**) ×400,000; and (**c**) ×800,000.

**Figure 9 micromachines-13-02009-f009:**
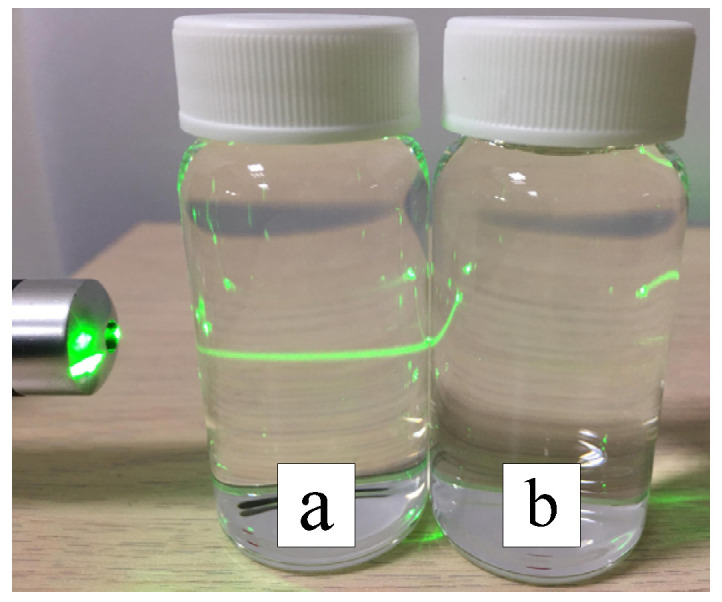
Colloidal solution of C-M-EDM 10-10-16 has a scattering trace of apparent laser light (**a**) Colloidal solution, (**b**) Deionized water.

**Figure 10 micromachines-13-02009-f010:**
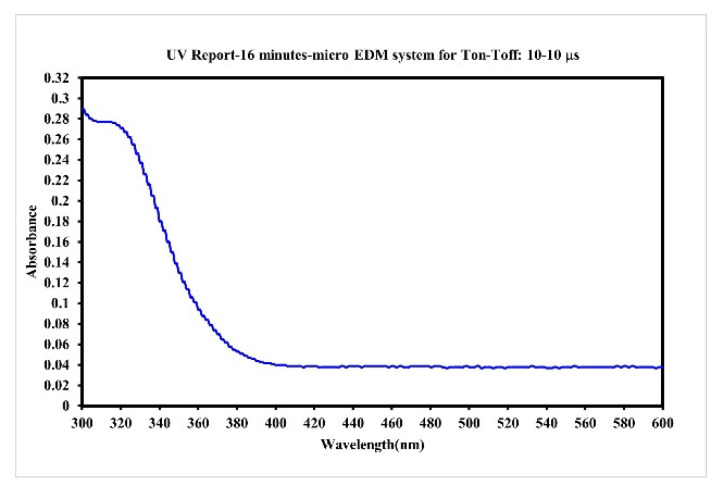
Absorption spectrum of the C-M-EDM 10-10-16 min colloid.

**Figure 11 micromachines-13-02009-f011:**
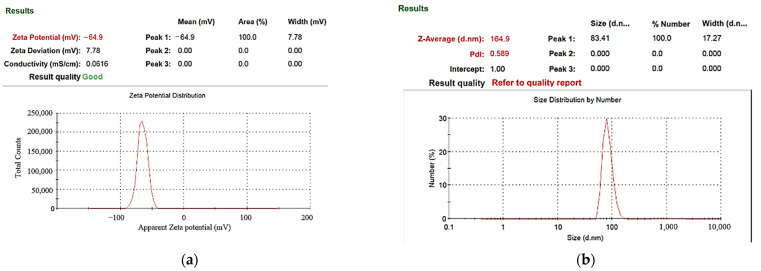
Analysis of C-M-EDM 10-10-16 min colloid: (**a**) ζ potential; (**b**) particle size distribution by number.

**Figure 12 micromachines-13-02009-f012:**
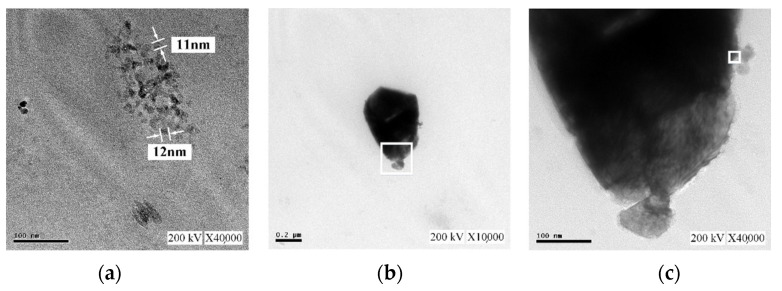
TEM images of the C-M-EDM 10-10-16 min colloid: (**a**) Site A under ×40,000 magnification; (**b**) Site B under ×10,000 magnification; (**c**) Site B under ×40,000 magnification.

**Figure 13 micromachines-13-02009-f013:**
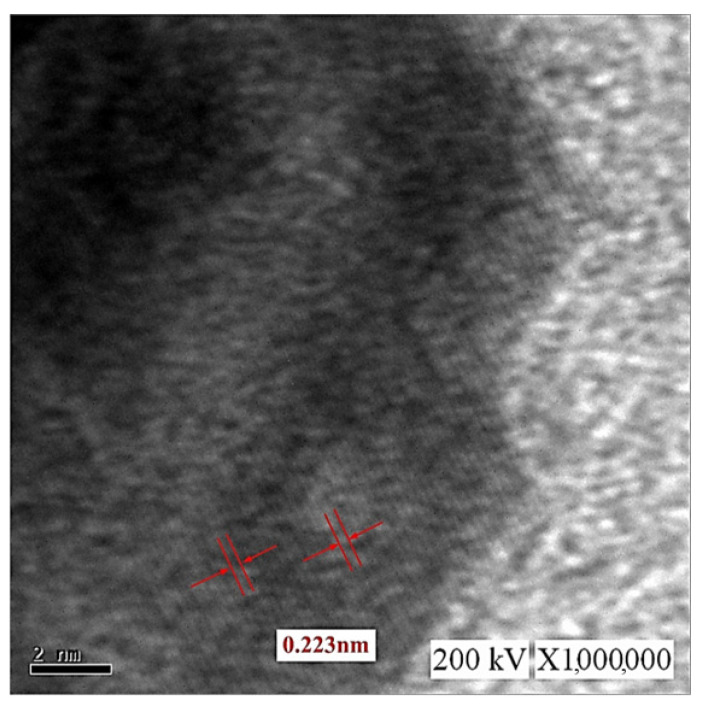
TEM images of the C-M-EDM 10-10-16 min colloid Site B under ×1,000,000 magnification.

**Figure 14 micromachines-13-02009-f014:**
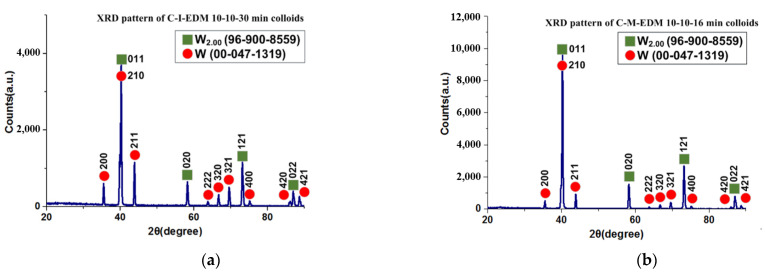
XRD pattern of the (**a**) C-I-EDM 10-10-30 min colloid and (**b**) C-M-EDM 10-10-16 min colloid.

**Table 1 micromachines-13-02009-t001:** Property comparison between the C-I-EDM 10-10-30 min colloid and C-I-EDM 30-30-30 min colloid.

Colloid	C-I-EDM 10-10-30 min Colloids	C-I-EDM 30-30-30 min Colloids
Wavelength	316 nm	315 nm
Absorbance peak	0.242	0.158
ζ potential	−61.3mV	−32.6mV
PDI	0.842	0.983
Average particle size	216.3 nm	252.2 nm
Particle size distribution at peak 1	80.51 nm	68.27 nm

**Table 2 micromachines-13-02009-t002:** Peak positions (2θ) and Miller indices for each component of the C-I-EDM 10-10-30 min colloid and C-M-EDM 10-10-16 min colloid.

Compound Name	Chem. Formula	The Positions 2θ and Miller Indices (h k l)
Tungsten	$W_{2.00}$	40.2° (0 1 1) 58.2° (0 2 0) 73.1° (1 2 1) 87.0° (0 2 2)
Tungsten	W	35.5° (2 0 0) 39.8° (2 1 0) 43.8° (2 1 1) 63.7° (2 2 2) 66.7° (3 2 0) 69.6° (3 2 1) 75.1° (4 0 0) 86.0° (4 2 0) 88.6° (4 2 1)

**Table 3 micromachines-13-02009-t003:** Property comparison of the C-I-EDM 10-10-30 min colloid and C-M-EDM 10-10-16 min colloid.

Colloid	C-I-EDM 10-10-30 min Colloids	C-M-EDM 10-10-16 min Colloids
Wavelength	316 nm	315 nm
Absorbance peak	0.242	0.277
Preparation time	30 min	16 min
ζ potential	−61.3 mV	−64.9 mV
Average particle size	216.3 nm	164.9 nm
PDI	0.842	0.589
Particle size distribution at peak 1	80.51 nm	83.41 nm
Lattice line width	0.223 nm	0.223 nm
Particle appearance and morphology	80 nm, quadrangle particle shape	11–12 nm, irregular particle shape

## Data Availability

The data used to support the findings of this study are included within the article.

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
