# Peer review of "A Study of Nano-Tungsten Colloid Preparing by the Electrical Spark Discharge Method"

_micromachines, 2022, doi:10.3390/mi13112009_

Round 1

Reviewer 1 Report

The paper describes a process for making colloidal nano-tungsten. As written, there are some issues that need to be addressed. Most notably, it is not clear what the new elements are in the process. This distinction is currently lost in Section 2. The authors need to state clearly and in simple terms how they are refining previous processes for making these nanomaterials.

1. Figure 1 does not use space efficiently. Are Figs. 2-4 really necessary? This is all review of previous setups and could simply be referenced. In general, Section 2 seems like it goes into a level of detail that might not need to be included, since the paper is already quite long.

2. Why is there an offset in the background of two absorption spectra in Fig. 8? Unless careful calibration based on concentration has been performed, it is not clear what absolute values of absorbance mean physically in this case (with a constant background). Care needs to be taken here

3. Figures 9, 10 and 13 are simply instrument readouts. It would be better to make an actual plot and list the parameters in a Table

4. Again, there is an unusual baseline in Fig. 12. What color are the suspensions? It would be useful to include a photograph of a suspension vial

5. I find the claim of lattice fringes in TEM to be unjustified, at least for the quality of images presented

6. What range of concentration are the colloids? From the TEM they are extremely sparse and dilute. It would be critical to know some sense of the yield obtained in the process.

Author Response

Dear reviewer, 

Please see the attachments for the response. Thank you.

Reviewer 2 Report

The study entitled Novel Design of the micro-EDM System for Preparing Nano-W Colloid is interesting and some novelity has done in the design of micro-EDM. Authors have presented the data in good format. The paper is quite well organized and its language is quite satisfactory. The paper is accepted with minor revision. However, there are some issues, which should be improved, and incorporated, in the revised manuscript.

Page 2: Line 57: Full form of ee-micro-EDM should be written in it’s first use.

Page 2: Line 71-74

T1, T2, T3, and T4 sequentially representing the discharge, ionization, slag generation and expulsion, and insulation recovery of the DF stages, respectively. T1–T3 occur during the Ton stage of the pulse period; T4 occurs during the Toff stage of the period.

Give more clarity on the above statement.

Page 6: Line 221

C-I-EDM 90-90-30 min means Ton 90 µs, Toff 90 µs and machining time is 30 minute. Is it right.

Page 7: Line 224-229

The absorbance decreases with the increase in pulse on time and pulse off time. What is the reason behind it. Incorporate in the revised manuscript.

Page 7: Line 230-234

ζ potential of 10-10-30 min colloid is higher than that of ζ potential of 30-30-30 min colloid. Give the reason.

Also the particles size obtained in 10-10-30 min colloid is less than that of 30-30-30 min colloid. Why is it so.

In Industrial EDM open circuit voltage and the peak current were set as 240 V and 6 A; however, during ee-micro EDM open circuit voltage and the peak current were set as 192 V and 4.7 A, respectively. Why you have chosen this parameter.

The time duration in ee-micro EDM is taken as 16 minutes whether in industrial EDM it is taken as 30 minutes. Justify this also.

Author Response

(The authors gave the same response as above.)

Reviewer 3 Report

This paper describes the nanoparticles of W using micro EDM. But there is no new knowledge to micro EDM technology. The description in conclusions is not correct. Those nanoparticles were generated at high temperature and high pressure given by electrical discharges.

Author Response

(The authors gave the same response as above.)

Reviewer 4 Report

This is an interesting study and the paper is generally well written and structured. However, in my opinion the paper has some shortcomings in regards to result discussion where the paper only reporting the finding without providing in depth discussion. I suggested citing more relevant and recent literature to support the finding.

Please explain why the preparation time of 16 min was chosen for C-M-EDM instead of 30 min? Is the comparison still valid if the processing parameters are not the same?

Author Response

(The authors gave the same response as above.)

Round 2

Reviewer 1 Report

The authors have not satisfactorily addressed my comments. In fact, they essentially ignored them, so I remain unenthusiastic about this manuscript. The issues that I cited in my previous report are still present.

Author Response

Dear reviewer, 

We have revised and replied to all the comments. Please check our modified version. Thank you.

Reviewer 3 Report

Electrical discharges can be used to generate micro/nano particles by the high temperature of plasma channels. It is possible to have nano particles of electrically conductive materials. There is no new knowledge to EDM technology.

Author Response

(The authors gave the same response as above.)

Round 3

Reviewer 3 Report

This paper describe experimental results of nano-W colloids by micro EDM. During EDM, the material is molten and partially vaporized, following by the debris which may be in micro and nano size. There is no new knowledge of EDM. This paper is not acceptable for publication.